# Ventilator Acquired Pneumonia in COVID-19 ICU Patients: A Retrospective Cohort Study during Pandemia in France

**DOI:** 10.3390/jcm12020421

**Published:** 2023-01-04

**Authors:** Jacques Moreno, Julien Carvelli, Audrey Lesaux, Mohamed Boucekine, David Tonon, Amandine Bichon, Marc Gainnier, Jeremy Bourenne

**Affiliations:** 1Intensive Care Unit, Timone University Hospital APHM, 13005 Marseille, France; 2Marseille Immunopole, School of Medicine La Timone Medical Campus, Aix-Marseille University, 13005 Marseille, France; 3EA 3279 CEReSS Health Service Research and Quality of Life Center, School of Medicine La Timone Medical Campus, Aix-Marseille University, 13005 Marseille, France; 4Support Unit for Clinical Research and Economic Evaluation, Department of Clinical Research and Innovation, Assistance Publique—Hôpitaux de Marseille, 13005 Marseille, France; 5Marseille Private Hospital, Beauregard Hospital, 13005 Marseille, France; 6INSERM, C2VN, School of Medicine La Timone Medical Campus, Aix-Marseille University, 13005 Marseille, France

**Keywords:** acute respiratory failure, ventilor acquired pneumonia, early antibio-therapy, infectious risk factors, SARS-CoV-2 pneumonia

## Abstract

Describe the characteristics of ventilation-acquired pneumonia (VAP) and potential risk factors in critically ill SARS-CoV-2 patients admitted in three French public hospitals during the first year of the COVID-19 pandemic. We conducted a monocentric retrospective study in seven Marseille intensive care units (ICUs) aiming to describe VAP characteristics and identify their risk factors. VAP patients were compared to a non-VAP control group. From March to November 2020, 161 patients admitted for viral-induced acute respiratory failure (ARF) requiring invasive mechanical ventilation (IMV) were included. This cohort was categorized in two groups according to the development or not of a VAP during their stay in ICU. 82 patients (51%) developed ventilation-acquired pneumonia. Most of them were men (77%) and 55% had hypertension. In the VAP population, 31 out of 82 patients (38%) had received dexamethasone and 47% were administered antibiotic course prior to ICU admission. An amount of 88% of respiratory infections were late VAPs with a median delay of 10 days from the onset of IMV. Gram negative bacteria were responsible for 62% of VAPs with *Pseudomonas* spp. being the most documented bacteria. Less than a third of the ICU-acquired infections were due to multidrug resistant (MDR) bacteria mainly displaying AmpC cephalosporin hyper production resistance phenotype. Multivariate analysis revealed that early Dexamethasone administration in ICU, male sex, older age and ROX score were risk factors for VAP whereas pre-ICU antimicrobial treatment and higher IGS 2 were protective factors. VAP is a frequent ICU-related complication affecting half of patients infected with SARS-CoV-2 and requiring IMV. It was responsible for increased morbidity due to a longer ICU and hospital stay. VAP risk factors included demographic factors such as age and sex. Dexamethasone was associated with a threefold greater risk of developing VAP during ICU stay. These results need to be comforted by large multi-centric studies before questioning the only available and effective treatment against SARS-CoV-2 in ICU patients.

## 1. Introduction

The COVID-19 pandemic has, since late December 2019, created a global health crisis, infected millions of people and caused a significant number of casualties [1]. Thanks to broad anti-viral management care including large-scale vaccination policies, Severe Acute Respiratory Syndrome Coronavirus 2-related (SARS-CoV-2) hospitalizations have decreased but its mutation-prone genome coupled with concerns about duration of vaccine effectiveness explain why it remains a public health concern to this day [2,3,4,5]. In 2020, France faced two waves of infection, the first in spring and the second in autumn, with a large number of patients admitted to the hospital and the intensive care units (ICU). This patient overflow saturated public hospital wards and critical care units, which in turn forced a rapid structural and labor force reorganization in order to maintain adequate health services. SARS-CoV-2-related symptoms vary from mild with fever, coughing, headaches and myalgia, to severe with life-threatening hypoxic respiratory failure [6]. In critically ill adult patients, severe acute respiratory syndrome (SARS) is the main cause of organ failure and the need for invasive mechanical ventilation (IMV) is often required [7,8]. The main complication of prolonged intubation is ventilator-acquired pneumonia (VAP). Increasing IMV duration and hospital length of stay [9,10], VAPs are responsible for significant morbidity and an estimated VAP-attributable mortality rates between 5–13% [11]. Although the incidence and mortality rates have decreased with the development of preventive strategies, VAP remains the most common cause of nosocomial infection in ICUs. Current challenges in the management of VAP include the lack of a gold standard for diagnosis, the absence of effective preventive strategies and the rise in microbial resistance [12]. SARS-CoV-2 patients are particularly at risk of developing VAP; indeed, several studies have shown that the average duration of mechanical ventilation is longer in COVID-19 Acute Respiratory Distress Syndrome (ARDS) compared to “classical” ARDS [13,14]. However, the incidence of nosocomial pneumonia in this population remains unclear with numbers ranging between 23% to 50% [15,16]. Risk factors for VAP have thoroughly been studied in the ICU population but there is little data regarding the COVID-19 ICU population. Dexamethasone has emerged as a cornerstone treatment for oxygen-requiring SARS-CoV-2 patients as it is the first drug having consistently shown decreased mortality [17]. On the other hand, corticosteroids have been associated with immunosuppression and subsequent superinfections in non-COVID ICU patients [18]. The impact of wide-use systemic corticosteroid therapy and immunomodulating drugs on VAP in a COVID-19 setting has yet to be fully understood. It stands to reason that further exploring the characteristics of VAP in a COVID-19 setting is the first step to reducing its incidence and improving the outcome of mechanically ventilated patients.

## 2. Materials and Methods

### 2.1. Setting and Study Design

This study was conducted to analyze the incidence and risk factors for VAP in COVID-19 patients during the first year of the pandemic in Marseille, France.

A retrospective analysis of prospectively collected data of 248 adult patients with severe COVID-19 admitted to seven ICUs of three tertiary public hospitals between March 2020 and November 2020 was performed. A number of 87 patients who were never exposed to invasive mechanical ventilation were excluded, leaving 161 mechanically ventilated patients left as our final cohort. We then distinguished 82 patients who developed at least one VAP during their ICU stay and proceeded to describe the incidence and characteristics of this group. The remaining 79 patients were used as a control group (Figure 1).

The participating centers shared comparable preventive care regarding hospital acquired infections including VAP. No selective digestive decontamination was used and stress ulcer prophylaxis was applied when indicated.

All consecutive patients with a SARS-CoV-2 infection diagnosis and admitted to the participating ICUs during the study period were included. Exclusion criteria encompassed age <18 years, admission to ICU for other reasons than COVID-19, and no exposure to invasive mechanical ventilation.

### 2.2. Diagnosis and Definitions

The COVID-19 infection diagnostis was obtained through polymerase chain reaction (PCR) performed on nasal or tracheo-bronchial swabs at admission. Nucleic acids were extracted using the EZ1 Virus Mini Kit v2.0 (Qiagen^®^, Courtaboeuf, France) and the two RT-PCR assays were carried out using the LightCycler Multiplex RNA Virus Master kit (Roche Diagnostics^®^, Mannheim, Germany) as previously described.

In a few cases with negative PCR swabs contrasting with strong clinical, radiological and anamnestic evidence, SARS-CoV-2 infection diagnosis was still diagnosed.

Regarding VAP in this study, the diagnosis was obtained with the combination of two criteria according to the latest European center for disease control’s definition for VAP [19]: (1) clinical, biological and radiological signs, leading the practitioner to suspect an infection and prescribe an antibiotic treatment, and (2) a positive respiratory microbiological isolate.

Infections were considered ICU-acquired infections if they occurred ≥48 h from ICU admission. VAP was classified into early versus late whether it occurred <5 days or ≥5 days from the start of mechanical ventilation.

### 2.3. Collected Data

The following patient data were collected at admission: age, sex, body mass index (BMI), comorbidities including diabetes, hypertension, cardiovascular disease, metabolic syndrome, active smoking, chronic lung disease, chronic kidney disease and immunocompromised status. Patient characteristics at ICU admission included severity scores, PaO_2_/FiO_2_ ratio and biological inflammation markers. Three critical patient severity scores were collected and are listed as follows: “*Index de Gravité Simplifié 2*” (IGS 2), Charlson and ROX. These complementary scores describe patient gravity, estimating predicted in-hospital mortality for the first two using patient characteristics within the first 24 h of ICU admission and patient comorbidities respectively. The ROX score is a tool predicting high-flow nasal cannula failure and the need for mechanical ventilation.

New prognostic biomarkers including neutrophil to lymphocyte ratio (NLR) and Systemic immune-inflammation index (SII) have been studied in recent years as potential tools for sepsis management [20] as well as COVID-19 prognosis [21] and were also collected for analysis.

Infected patients within the first 48 h of ICU admission were also reported.

We also described infectious-related therapies used before and during early ICU stay such as corticosteroids, immunomodulating drugs (anti IL-6, anti IL-1, ruxolitinib, Lopinavir/Ritonavir), hydroxychloroquine and antibiotics.

ICU patient management, complications and outcomes were collected. Respiratory support such as high-flow nasal cannula (HFNC) and non-invasive ventilation (NIV) use, prone positioning, use of neuromuscular blockers (NMB), the need for Extracorporeal membrane oxygenation (ECMO) as well as high-dose corticosteroid therapy were reported. Other organ failures and support systems such as septic shock and norepinephrine use, acute renal failure and renal replacement therapy (RRT) were documented. Median mechanical ventilation duration, hospital and ICU length of stay (LOS) and mortality rates were also reported.

### 2.4. Microbiological Investigations

The bacteriologic diagnosis was obtained by quantitative culture using a positive threshold of 10^5^ and 10^2^ Colony Forming Units (CFU/mL) for endotracheal aspirates and BAL fluids, respectively. Cultures were considered as polymicrobial if >2 microorganisms grew. Isolates were identified using Matrix-assisted laser desorption ionization-time of flight mass spectrometry (MALDI-TOF MS) with a log score >1.9 as previously described. Antibiotic susceptibility testing was performed using disk diffusion method according to the European Committee on Antimicrobial Susceptibility Testing (EUCAST) and minimum inhibitory concentrations (MICs) were obtained using the E-test method (Biomérieux, Marcy l’Etoile, France).

Among enterobacteria, extended spectrum beta-lactamase producers (ESBL) were identified when a synergy between ceftriaxone and clavulanate was observed. Other enterobacteria resistant to ceftriaxone were considered as AmpC-hyper producers (CASEH). We also screened for methicillin-resistant *Staphylococcus aureus* (MRSA) and *Stenotrophomonas maltophilia* resistant isolates.

*Herpesviridae* (Herpes simplex virus, cytomegalovirus and Epstein-Barr virus) were reported but were considered to be reactivations and not considered a component of VAP. Viral reactivations (HSV and CMV) and median Ct SARS-CoV-2 viral loads and duration of PCR positivity were also documented.

We also screened for evidence of invasive pulmonary aspergillosis (IPA), as described in COVID-19 patients [22].

### 2.5. Statistical Analysis

Descriptive statistics were produced for demographic, clinical, and laboratory characteristics of patients. Mean, median and interquartile range [IQR] are reported for continuous variables, and numbers and percentages are reported for categorical variables. Groups were compared with unpaired *t*-test or Mann-Whitney nonparametric tests, according to data distribution, for continuous variables and with chi-square test (or Fisher exact test when appropriate) for categorical variables.

The most relevant risk factors associated with VAP were identified by univariate logistic regression analysis. Variables with a *p*-value of less than 0.2 in univariate analysis were included in the final multivariable logistic regression model. Odds ratios (ORs) and 95% CIs are presented. SPSS version 20.0 software (IBM, Armonk, NY, USA) was used for all statistical analyses. A *p*-value of < 0.05 was regarded as statistically significant.

## 3. Results

### 3.1. Patient Characteristics

82/161 (51%) patients developed at least one ventilation-acquired pneumonia during ICU stay.

VAP patient clinical characteristics are summarized in Table 1. Median age was significantly slightly higher in this population at 65 years compared to 62 years in the non-VAP group (*p* = 0.02). An amount of 83% were men and the average BMI was 27.7 kg/m^2^ (25.3–32.1) with 32% of obese patients. An amount of 91% of VAP patients displayed at least one comorbidity, the most frequent being high blood pressure and diabetes, present in 45/82 patients (55%) and 25/82 patients (30%) respectively. Comorbidities were similar in VAP and non-VAP groups. Active smoking was rare in our study, reported at 6%.

Table 1 also reported patient status at ICU admission. Within the three severity scores, the ROX predictor scored significantly worse in the ventilation-acquired pneumoniae cohort compared to the control group (3.7 vs. 5.8; *p* = 0.012). Biological markers such as ferritin, fibrinogen, D-dimer levels were similar in both groups; however, C-reactive protein serum levels were higher in VAP patients (185 mg/L vs. 98 mg/L; *p* = 0.03). Both NLR and SII biological parameters were significantly higher in the VAP group (*p* = 0.016). A total of 13/82 patients were infected within the first 48 h of ICU admission (16%), with an equal percentage found in the non-VAP group. Percentage of CT scan lung lesions at ICU admission were also statistically more important in VAP patients with an average of 37% of damaged lung compared to 27% in the control group (*p* = 0.038). Delay between conventional hospitalization and ICU admission was short, with patients being transferred to the ICU at day 0 in the VAP group versus day 1 in non-VAP patients; however, this difference was not statistically significant.

### 3.2. Infectious Disease Management

Table 2 summarizes antimicrobial courses administered before and during ICU stay. Half of VAP patients (47%) had received antibiotics before ICU admission, significantly less than non-VAP patients (65%; *p* = 0.024), with a median treatment duration of 3 days before ICU (IQR, 2-5). Hydroxychloroquine use was similar in both groups, prescribed in about 30% of patients in pre-ICU and early ICU settings. Rate of antibiotic in the first 2 days of ICU stay reached 83% in VAP patients, with broad spectrum antibiotics predom- inantly used (i.e., 3rd generation cephalosporins, fluoroquinolones, *β* -lactam and *β* -lactam inhibitors, carbapenems, aminosides, anti-methicillin-resistant *Staphylococcus aureus* [MRSA]). Dexamethasone 6 mg per day was used in 31/82 (38%) VAP patients and 20/79 (25%) non-VAP patients (*p* = 0.09) with a global incidence of 31% when analyzing our 161 ventilated patients. Immunomodulatory drugs were administered in 33% of VAP patients and in 21% of non-VAP patients (*p* = 0.1).

### 3.3. Patient Management, Complications and Outcome during ICU Stay

Information about patient respiratory management and other ICU specific therapies as well as complications are detailed in Table 3. Noninvasive respiratory support consisted in high-flow nasal cannula and noninvasive ventilation, used respectively in 63% and 28% of VAP patients, respectively. There was no difference in the usage rate of use of these treatments between the two groups. Over 90% of patients, regardless of the group, required neuromuscular blockers and 13/82 VAP patients (16%) required extra corporeal membrane oxygenation (ECMO) support. Nitric oxyde was used twice as often in the VAP group than in the control group (*p* = 0.004). High-dose methylprednisolone treatment was statistically associated with VAPs (*p* = 0.006). Although norepinephrine was more frequently used in VAP patients (*p* = 0.001), there was no difference in incidence of septic shock or acute renal failure between both groups. Pneumothorax and cardiac rhythm disorders were also more frequently reported in VAP patients. Duration of COVID-19 PCR positivity was 13 days in the VAP group and 11 days in the non-VAP group (*p* = 0.2). ICU LOS, hospital LOS and days under mechanical ventilation were all significantly longer in VAP patients (*p* < 0.001). A 25% mortality rate was reported and did not significantly differ between groups.

### 3.4. VAP Characteristics and Bacterial Documentation

The type and incidence of each bacterium responsible for VAPs is detailed in Table 4 and Figure 2. These infections were categorized into early and late VAPs, on whether they occurred before or after the 5th day of mechanical ventilation. An amount of 72 patients (88%) developed late VAPs with a median delay of 10 days after intubation (IQR; 7–17). A total of 31% of VAPs were due to multidrug resistant bacteria (MDR) with no significant difference between early and late infections. VAP associated blood stream infections (BSI) were exclusively documented in late VAPs (20/72), with 33% caused by MDR bacteria.

Regardless of the time to onset of VAP, three organisms were responsible for 60% of infections: *Klebsiella* spp., *Pseudomonas* spp. and *Staphylococcus aureus*. Gram negative species were predominantly involved in 62% of VAPs. When analyzing the different types of bacteria between both VAP groups, we noticed that *Staphylococcus aureus* was mainly documented in late VAPs with an incidence of 22% vs. only 6% in early VAPs, however this difference was not statistically significant (*p* = 0.08). Most of the strains were Methicillin-sensitive *Staphylococcus aureus* (MSSA) while only 5/29 isolates displayed a Methicillin-resistant phenotype (MRSA), the latter all being reported in late VAPs. There were no differences between both groups regarding other bacterial species involved in VAPs except from *Acromobacter xylosoxidans* which was mainly found in early VAPs (*p* = 0.005). There were no recorded cases of invasive pulmonary aspergillosis in this study.

### 3.5. Antibiotic Resistance Mechanisms

Finally, we analyzed the different types of antibiotic resistance mechanisms among VAP and BSI MDR infections (Figure 3). These infections regarded 26% of patients who experienced at least one infection during their ICU stay and in 17% of our ventilated cohort. The dominant resistance mechanisms were cephalosporinase hyper-production (CASEH), Ceftazidime resistant *Pseudomonas aeruginosa* (CAZ I/R PA) and extended-spectrum beta-lactamase (ESBL) with respective incidences of 41%, 24% and 15%. MRSA was documented in 6/46 isolates. Early VAPs and VAP-free blood stream infections accounted for 4/46 of BMR infections (9%) in ventilated patients, whilst late VAPs accounted for 67%.

*Klebsiella* spp. (*pneumoniae, aerogenes*) was the main strain displaying CASEH and ESBL resistance mechanisms with incidences of 58% and 86%, respectively. Other *enterobacteriaceae* species involved included *Enterobacter cloacae*, *Escherichia coli* and *Hafnia alvei*.

### 3.6. Multivariate Analysis

In this multivariate analysis (Table 5), we singled out clinically relevant factors that had a univariate *p* score < 0.2 in order to identify potential VAP risk factors. Age and male sex were associated to VAP (*p* = 0.014 and 0.018, respectively). IGS 2 and ROX scores at admission were associated to VAP with paradoxically higher IGS 2 scores having a protective effect. Dexamethasone was significantly associated with VAP incidence OR 3.2 (1.01–10), *p* = 0.046. In univariate analysis, antimicrobial treatment prior to ICU admission was strongly correlated with lower VAP incidence, thus constituting a protective fac-tor [*p* = 0.009; OR = 0.23 (0.07–0.7)]. Neutrophil to lymphocyte ratio was no longer associated to VAP in multivariate analysis.

## 4. Discussion

### 4.1. Ventilation-Acquired Pneumonia Characteristics

The studied population was mainly composed by comorbid males (83%) with high blood pressure and diabetes being the two most described comorbidities. This reinforces the already common knowledge that SARS-CoV-2 is most virulent in these types of patients [23]. Time between ICU admission to intubation was very short with a median intubation date at day 1 reflecting the critical level of respiratory failure these patients faced.

Our work provides a detailed description of VAP microbiological distribution. Infections occurred at day 3 (IQR, 3–3) for early VAP and day 9 (6–14) for late VAPs. The proportion between early and late onset VAPs in this study, with a majority of late VAPs (88%) is consistent with other reports. Most infections were caused by Gram negative bacteria (62%) with *Pseudomonas aeruginosa* being the most documented bacteria (21%). These numbers are similar to the ones published by Grasselli et al. [24]. *Staphylococcus aureus* was mostly found in late VAPs (*p* = 0.08) contrasting with the distribution of Gram-positive bacteria in VAP settings that are usually found in the early stages. *Staphylococcus aureus* has demonstrated the ability to interact and infect different cell strains, while becoming increasingly resistant to antibiotic therapy and a reservoir of bacteria that can make the infection difficult to treat [25]. However, this could simply be explained by a lack of power from low early VAP incidence.

Another key finding with potentially significant clinical implications is that MDR bacteria caused about one third of VAPs. The most common antibiotic resistance mechanism being AmpC cephalosporin hyperproduction (CASEH) found in 90% of the cases in either *Klebsiella* or *Enterobacter* spp. MRSA strains represented 17% of all *Staphylococcus aureus* VAPs, remaining low and consistent with literature. This high proportion of MDR infections could be explained by a large amount of patients (47%) receiving broad spectrum antibiotics prior to ICU admission, including 3rd generation cephalosporins. Additionally, the influx of critically ill patients during the pandemic may have influenced the quality of critical care provided by overloading medical staff, potentially increasing the risk of ventilation-associated lower respiratory tract infections (VA-LRTI) by MDR bacteria. Indeed, previous studies reported that contact isolation measures, especially inappropriate glove use, could increase the transmission of MDR bacteria [26].

### 4.2. Ventilation-Acquired Pneumonia Risk Factors

Age and sex where the two demographic characteristics associated with VAPs in multivariate analysis. Age has widely been recognized as a risk factor for nosocomial infection and VAP in COVID and non-COVID settings. Various mechanisms such as decreased immune function, frequent chronic illnesses and malnutrition explain why older patients are more likely to develop hospital-acquired infections such as VAPs [24,27]. Male gender has been associated with VAP in non-COVID patients [28] but our study is the first to report this risk factor in a COVID cohort. However, the SARS-CoV-2 itself could be a confounding factor as we know that this virus is more virulent in men and responsible for higher ICU admission and mortality rates compared to women [29].

The ROX score, recently developed, has been used during the COVID-19 pandemic by certain teams to predict the need for invasive mechanical ventilation after high-flow nasal cannula failure [30]. This clinical score (SpO2/FiO2/RR) was the only score associated to VAP incidence and may reflect that the level of acute respiratory failure before intubation could be associated to the development of VA-LRTI.

The role of dexamethasone’s impact on VAP was a key question in this study. This treatment at a dose of 6 mg per day during 10 days is a cornerstone treatment for severe cases of COVID-19 infection, as it is the only treatment, being associated with a significant lower in-ICU mortality [17]. However, the impact of this treatment on VAP incidence in SARS-CoV-2 infected patients is still up for debate with contrasting literature on the subject. Reys et al. revealed in a prospective study that dexamethasone was a risk factor for ICU-acquired respiratory tract infections whilst Gragueb-Chatti et al. found no relation between VAP or BSI and dexamethasone [31,32]. In our study, the multivariate analysis revealed that dexamethasone was associated with higher VAP incidence with an odds ratio of 3.2 (1.01–10). Other corticosteroid treatments were used such as high-dose methylprednisolone which had a higher incidence in the VAP group. However, this treatment was mainly started after the diagnosis of VAP making its causality as a potential risk factor difficult to judge. Dexamethasone’s purpose is to mitigate inflammation-related organ injuries. ICU patients with low biological markers at admission could be a sub-group of patients in which dexamethasone favors bacterial super infections rather than treating inflammation-induced lesions. Further prospective studies are needed to confirm this potential unintended effect. In this situation, alveolar injury marker can be useful for the diagnosis of VAP. Recently Angiopoietin-2 (Ang-2) and soluble intercellular adhesion molecule-1 (ICAM-1), were identified as markers of pulmonary endothelial injury in COVID-19 ARDS [33].

The multivariate analysis revealed that antibiotic treatment administered prior to ICU admission was associated with a reduction in VAP incidence (see Table 5). Based on our definition of VAP, high antibiotic exposure prior to bacterial samplings might have caused falsely negative results leading to unaccounted non-documented VAPs. Respiratory bacterial co-infection rates during standard hospitalization in COVID-19 were low, with 755 reported in literature [34,35]. The impact of inadequate probabilistic antibiotic treatment on the emergence of antibiotic resistance has been an ongoing and growing global health issue [12]. Coupling this to the low evidence level of results obtained from our retrospective study, it seems unreasonable to advocate for pre-emptive antibiotic treatment before ICU admission with the objective of reducing VAP incidence in COVID-19. On the other hand, recent ICU health care-associated pneumonia (HCAP) recommendations stated that systemic prophylactic antibiotics, through selective digestive decontamination, may help reduce HCAP incidence [36]. Future prospective studies are needed to clarify the use and timing of prophylactic antibiotics. Recently, multivalent fucose derivative has been described potential broad spectrum ant biofilm agent, although its place in adjunctive treatment of VAP needs to be clarified [37].

Neutrophil to lymphocyte ratio at ICU admission was associated to VAP in the univariate analysis but no longer in the multivariate analysis. NLR has been used in these last years as a marker for endothelial dysfunction, a mechanism particularly present in COVID-19 pathogenesis. Different studies found associations between this biological marker and poor clinical outcome, with Jimeno et al. demonstrating a relation between peak NLR and death. However, NLR lacks specificity and was mostly described in prognostication of sepsis and oncological diseases making it difficult to use on a practical basis. Disease-specific thresholds may increase the reliability of this marker and could be a focus point for future studies.

### 4.3. Incidence of Ventilation-Acquired Pneumonia

In this retrospective study, we analyzed the epidemiologic and etiological factors influencing the development of VAP in a cohort of critically ill COVID-19 patients. The incidence of VAP in this study was 51% as reported in literature with incidences ranging between 44% to 64% [38,39]. This contrasts with VAP occurrence in non-COVID ICU settings which seem to have lower incidence rates (13–29%) [16,28]. One of the recurrent explanations mentioned in literature is the longer duration of mechanical ventilation in SARS-CoV-2-related ARDS compared to other types of ARDS. In our study, the median duration of mechanical ventilation was 18 days (IQR, 8.2–33) which is slightly longer than other reports (13–15 days). This could be due to the fact that this study analyzes the earliest period of the COVID-19 pandemic, a time in which intensivists were encouraged to intubate patients at early stages of respiratory failure. Besides, the tropism of SARS-CoV-2 for nervous tissue and its association with confusion agitation and encephalopathy [40], may lead to a more difficult to weaning from mechanical ventilation. On a cellular level, one of the strongest predictors of nosocomial infection in critically ill patients is impaired immune cell function. Patients with COVID-19 experience a complex dysregulation of their immune function with features of both hyper-inflammatory activation and organ damage as well as impaired antimicrobial functions. Notably, damage to the alveolar membrane, although not specific to COVID-19, may facilitate invasion of bacterial species [41], similarly to influenza infection. However, when comparing VAP incidences in a COVID versus influenza ICU population, these remained significantly higher in the COVID-19 cohort (50% vs. 30%) [42]. A reason for this could be widespread thrombosis which distinguishes the pulmonary pathophysiology of COVID-19, from that of equally severe influenza virus infection. These lesions might promote local immunity alteration, bacterial colonization, and further lung infection. However, further studies are required to confirm this hypothesis.

### 4.4. Limitations

The retrospective design of our study could lead to an interpretative bias. The low incidence of dexamethasone is not representative of an up-to-date ICU COVID-19 cohort and weakens the generalization of our results. Blood stream infections were mainly associated with VAP (80%), thus the analysis of their association with dexamethasone was difficult.

## 5. Conclusions

In this retrospective study, we analyzed the characteristics and risk factors involved in ventilation-acquired pneumonia in a COVID-19 cohort within three French public hospitals from March to November 2020. After adjusting for confounding factors, the main risk factors for ventilation-acquired pneumonia were dexamethasone, older age and male gender whereas antibiotic treatment prior to ICU admission was a protective factor. As dexamethasone has been the only evidence-based treatment that has shown decreased mortality in severe cases of COVID-19, future robust multicentric studies are needed before questioning the risk-benefit ratio of this treatment.

## Figures and Tables

**Figure 1 jcm-12-00421-f001:**
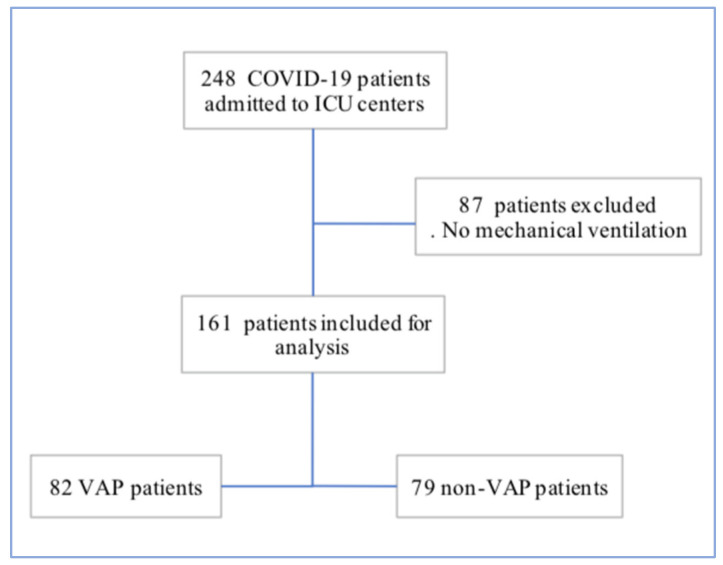
Study Flow chart.

**Figure 2 jcm-12-00421-f002:**
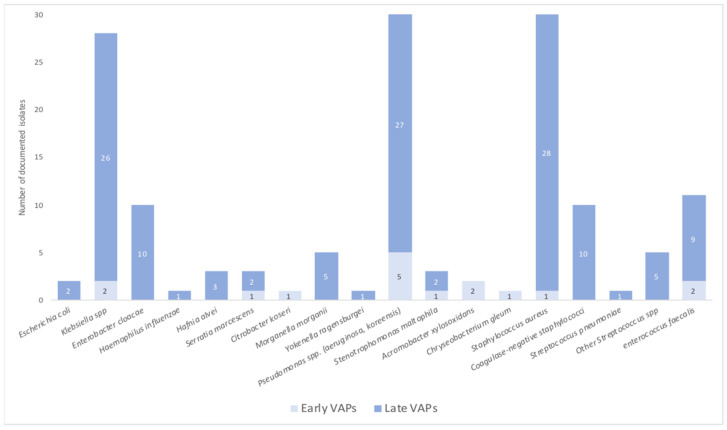
Ventilation-acquired pneumonia bacterial documentation.

**Figure 3 jcm-12-00421-f003:**
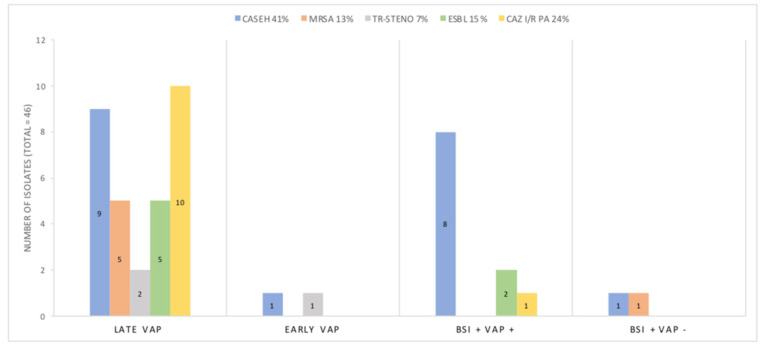
Antibiotic resistance mechanisms according to type of infection and bacteria.

**Table 1 jcm-12-00421-t001:** Patient clinical and biological characteristics.

Variable	All Patients		VAP Patients		Non VAP Patients		*p*-Value	OR (95% CI)
	N = 161		N = 82		N = 79			
			(51%)		(49%)			
Age—YEAR, median	63	(57–72)	65	(58–73)	62	(54–69)	0.02	1.03 (1.00–1.06)
Sex—Male	124	(77)	68	(82.9)	56	(70.9)	0.069	
BMI kg/m^2^, median	28	(25–32)	27.7	(25.3–32.1)	28	(25.8–33.8)	0.51	0.98 (0.93–1.03)
18–24.9	31	(19)	16	(23.2)	15	(21.4)	0.8	
25–29.9	52	(32)	27	(39.1)	25	(35.7)	0.67	
30–40	49	(30)	23	(33.3)	26	(37.1)	0.63	
>40	7	(4)	3	(3.9)	4	(5.3)	0.71	
Comorbidities	147	(91.3)	75	(91.5)	72	(91.1)		
Hypertension	88	(54)	45	(54.9)	43	(54.4)	0.95	1.01 (0.54–1.89)
Diabetes	58	(36)	25	(30.5)	33	(41.8)	0.13	0.61 (0.32–1.17)
Cardiovascular disease	31	(19)	19	(23.2)	12	(15.2)	0.2	1.68 (0.75–3.74)
Coronary disease	20	(12.4)	11	(13.4)	9	(11.4)	0.69	
Chronic heart failure	3	(19)	1	(1.2)	2	(2.5)	0.61	
Occlusive arteriopathy	4	(2.5)	3	(3.7)	1	(1.3)	0.62	
Metabolic syndrome	36	(22.4)	16	(19.5)	20	(25.3)	0.37	
Current smoking	6	(3.7)	5	(6.1)	1	(1.3)	0.14	5.06 (0.57–44.3)
Chronic lung disease	39	(24)	20	(24.4)	19	(24.1)	0.96	1.01 (0.49–2.09)
Obstructive sleep apnea	21	(13)	11	(13.4)	10	(12.7)	0.88	
Chronic kidney disease	12	(7.5)	4	(4.9)	8	(10.1)	0.21	0.45 (0.13–1.57)
Immunocompromised *	20	(12.4)	11	(13.4)	9	(11.4)	0.69	1.2 (0.47–3.08)
Chronic treatment	103	(64)	52	(63.4)	51	(64.6)	0.88	0.95 (0.5–1.81)
Status at ICU admission								
Delay between hospitalisation and ICU, d	0	(1–3)	0	(0–1.25)	1	(0–4)	0.12	
IGS 2	35	(28–43)	34	(30–43)	35	(27–43)	0.86	
CHARLSON	3	(1–4)	3	(2–4)	2	(1–4)	0.1	
SOFA	4	(3–7)	4	(3–7)	4	(2–7)	0.43	
ROX	4.6	(3–9.3)	3.7	(3–7)	5.8	(3–10.1)	0.012	
PaO_2_/FiO_2_ ratio (mmHg)	131	(96–180)	132.5	(91.5–179)	130	(100–181.5)	0.42	0.99 (0.99–1)
Maximum Temperature (°C)	38.3	(37.5–39)	38.3	(37.6–38.7)	38.4	(37.3–39)	0.7	0.93 (0.67–1.31)
Neutrophil to lymphocyte ratio	7.3	(5.1–13)	8.3	(5.5–15.6)	6	(4.6–9.2)	0.016	
Systemic immune-inflammation index (SII)	1610	(971–3270)	2113	(1116–3900)	1281	(834–2054)	0.016	1.11 (1.01–1.2)
Ferritin (µg/L)	1427	(1013–2345)	1241	(796–1611)	2273	(976–4141)	0.19	
Fibrinogen (g/L)	6.6	(5.9–7.7)	6.7	(5.9–8)	6.6	(5.7–7.3)	0.59	
D-dimer (µg/L)	1.5	(1–4)	1.5	(0.8–4)	1.6	(1.1–2.9)	0.9	
Procalcitonin (ng/mL)	0.3	(0.19–0.58)	0.3	(0.2–0.5)	0.3	(0.2–1.3)	0.49	
C-reactive protein (mg/dL)	170	(91.6–227)	185.9	(120–251)	98.7	(83.6–184)	0.03	0.89 (0.64–1.23)
Infection within 48 h of ICU admission	26	(16.1)	13	(15.9)	13	(16.5)	0.91	
CT scan lung lesion (%)	31.2		37.3		27.8		0.038	

Data throughout are presented as No. (%) of the included patients or median (interquartile range). *: includes solid tumors, active hematological malignancy, solid organ transplant, HIV, immunosupressive treatment.

**Table 2 jcm-12-00421-t002:** Anti-infective therapies.

Variable	All Patients		VAP Patients		Non VAP Patients		*p*-Value	OR (95% CI)
	N = 161		N = 82		N = 79			
			(51%)		(49%)			
Early ICU management								
Antibiotic treatment within 48 h of ICU admission	137	(85.6)	68	(82.9)	69	(88.5)	0.32	
Corticosteroids								
Dexamethasone 6 mg	51	(31.7)	31	(37.8)	20	(25.3)	0.09	1.8 (0.9–3.5)
Median duration of dexamethasone 6 mg	10	(7–10)	10	(7–10)	10	(6–10)	0.22	
Immunomodulators	44	(27.3)	27	(32.9)	17	(21.5)	0.1	
Anakinra (anti IL-1)	19	(11.8)	10	(12.2)	9	(11.4)	0.87	
Tocilizumab (anti IL-6)	4	(2.5)	1	(1.2)	3	(3.8)	0.31	
Ruxolitinib (Jakavi)	19	(11.8)	12	(14.6)	7	(8.9)	0.26	
Lopinavir/Ritonavir	18	(11.2)	13	(15.9)	5	(6.3)	0.063	
Hydroxychloroquine	107	(66.5)	50	(61)	57	(72)	0.13	
Pre ICU management								
Antibiotic treatment	90	(56.3)	39	(47.6)	51	(65.4)	0.024	0.48 (0.25–0.91)
3rd generation cephalosporin	60	(37.2)	25	(30)	35	(44)	0.069	
Azithromycin	60	(37.2)	24	(29)	36	(45.5)	0.035	
Penicillin A +/− Clavulanic acid	20	(12.4)	9	(10.9)	11	(13.9)	0.57	
Piperacillin—Tazobactam	11	(6.8)	6	(7.3)	5	(6.3)	0.8	
Duration of pre ICU antibiotic treatment, d (IQR)	4	(2–5)	3	(2–5)	4	(2–5.5)	0.16	
Hydroxychloroquine	52	(32)	23	(28)	29	(36.7)	0.24	
Dexamethasone 6 mg	8	(5)	3	(3)	5	(6.3)	0.48	

**Table 3 jcm-12-00421-t003:** ICU management and outcome.

Variable	All Patients		VAP Patients		Non VAP Patients		*p*-Value	OR (95% CI)
	N = 161		N = 82		N = 79			
			(51%)		(49%)			
Respiratory management								
Treated with High-flow nasal cannula (HFNC)	105	(65.2)	52	(63.4)	53	(67.1)	0.62	0.85 (0.44–1.6)
Treated with Non-invasive ventilation (NIV)	43	(26.7)	23	(28)	20	(25.3)	0.69	1.15 (0.57–2.3)
Neuromuscular blockers (NMB)	152	(94.4)	79	(96.3)	73	(92.4)	0.32	
Prone position	132	(82)	72	(87.8)	60	(75.9)	0.05	
Nitric oxyde (NO)	52	(32)	35	(42.6)	17	(21.5)	0.004	
Extracorporeal membrane oxygenation (ECMO)	24	(14.9)	13	(15.9)	11	(13.9)	0.73	
Solumedrol 2 mg/kg	21	(13)	17	(20.7)	4	(5.1)	0.006	4.9 (1.5–15.3)
ICU complications and outcome								
Norephinephrin > 1 mg/h	82	(50.9)	52	(63.4)	30	(37.9)	0.001	
Septic shock	50	(31)	30	(36.6)	20	(25.3)	0.12	
Blood stream infections	40	(25.5)	32	(80)	8	(20)	< 0.001	
Acute renal failure	76	(47.5)	43	(52.4)	33	(42.3)	0.2	
Renal replacement therapy	24	(14.9)	13	(15.9)	11	(13.9)	0.73	
Pneumothorax	11	(6.9)	10	(12.2)	1	(1.3)	0.006	
Rythm disorders	39	(24.2)	30	(36.6)	9	(11.4)	< 0.001	
Conduction disorders	14	(8.7)	10	(12.2)	4	(5.1)	0.108	
HSV-1 reactivation	19	(11.8)	13	(15.9)	6	(7.6)	0.104	
CMV reactivation	18	(11.2)	13	(15.9)	5	(6.3)	0.055	
SARS-CoV 2 viral loads (Ct/mL)	28.9	(24.9–31.7)	29.7	(24.8–32)	28	(24.9–31)	0.38	1.03 (0.96–1.10)
Duration of PCR positivity, d	12	(6.5–17)	13	(8–17)	11	(5.7–17)	0.2	1.02 (0.98–1.07)
Delay between ICU and IMV	0	(0–1.5)	0	(0–2)	1	(0–1)	0.2	
IMV duration, d	18	(8.2–33)	31.5	(15–47.2)	10	(5–18.2)	< 0.001	
Length of stay—ICU, d	24.5	(15–43)	39.5	(21–57)	17	(8–25)	< 0.001	
Length of stay—Hospital, d	35	(21–50)	43	(30–62.5)	24	(14–37.5)	< 0.001	
Death during ICU stay	41	(25.5)	21	(25.6)	20	(25.3)	0.96	

**Table 4 jcm-12-00421-t004:** Ventilation-acquired pneumonia characteristics and bacterial documentation.

Variable	All VAPs		Early VAPs		Late VAPs		*p*-Value
	N = 82		N = 10		N = 72		
Average delay of VAPs, d	9	(6–14)	3	(3–3)	10	(7–17)	< 0.001
Polymicrobial VAPs (> 2 isolates)	15	(18)	2	(20)	13	(18)	1
Multi-drug resistant (MDR) VAPs	26	(31.7)	2	(20)	24	(33)	0.49
VAP associated blood stream infections (BSI)	20	(24)	-		20	(27.7)	
VAP associated MDR BSIs	8	(33)			8	(33)	
Documented microbiological isolates	N = 148		N = 16		N = 132		
Gram negative bacteria (%)	(62)		(81)		(59)		
Fermenters							
*Escherichia coli*	2				2	(2.7)	1
*Klebsiella* spp.	28	(18.7)	2	(12)	26	(19)	0.87
*Enterobacter cloacae*	10	(6.7)			10	(7)	0.45
*Haemophilus influenzae*	1				1	(0.7)	1
*Hafnia alvei*	3				3	(2.2)	1
*Serratia marcescens*	3		1	(6)	2	(2.7)	0.8
*Citrobacter koseri*	1		1	(6)			0.24
*Morganella morganii*	5				5	(3.7)	0.87
*Yokenella ragensburgei*	1				1	(0.7)	1
Non fermenters							
*Pseudomonas* spp. (*aeruginosa*, *koreensis*)	32	(21.4)	5	(31)	27	(20)	0.67
*Stenotrophomonas maltophila*	3		1	(10)	2	(2.7)	0.8
*Acromobacter xylosoxidans*	2		2	(12)			0.005
*Chryseobacterium gleum*	1		1	(6)			0.24
Gram positive bacteria (%)	(38)		(19)		(41)		
*Staphylococcus aureus*	29	(20)	1	(6)	29	(22)	0.08
Methicillin-sensitive *Staphylococcus aureus*	24		1		23		0.68
Methicillin-resistant *Staphylococcus aureus*	5				5		0.76
Coagulase-negative *staphylococci*	10	(6.7)			10	(7)	0.45
*Streptococcus pneumoniae*	1				1	(0.7)	1
Other *Streptococcus* spp.	5				5	(3.7)	1
*Enterococcus faecalis*	11	(7.4)	2	(12)	9	(6.7)	0.87

**Table 5 jcm-12-00421-t005:** Multivariate risk factors associated with VAPs.

Variable	VAP Patients		Non VAP Patients		OR (95% CI)	Multivariate
	N = 82		N = 79			Adjusted
	(51%)		(49%)			*p*-Value
Age—YEAR, median	65	(58–73)	62	(54–69)	1.08 (1.01–1.15)	0.014
Sex—Male	34	(30–43)	35	(27–43)	3.99 (1.26–12.6)	0.018
IGS 2	34	(30–43)	35	(27–43)	0.94 (0.89–0.99)	0.037
CHARLSON	3	(2–4)	2	(1–4)	1.19 (0.79–1.78)	0.39
ROX	3.7	(3–7)	5.8	(3–10.1)	0.85 (0.75–0.97)	0.015
Dexamethasone 6 mg	31	(37.8)	20	(25.3)	3.2 (1.01–10)	0.046
Immunomodulatory therapy	27	(32.9)	17	(21.5)	2.73 (0.84–8.8)	0.092
Antibiotic course prior to ICU	39	(47.6)	51	(65.4)	0.23 (0.07–0.7)	0.009
Neutrophil Lymphocyte ratio	8.3	(5.5–15.6)	6	(4.6–9.2)	0.99 (0.95–1.03)	0.82
Duration of PCR positivity, d	13	(8–17)	11	(5.7–17)	1.02 (0.96–1.08)	0.41
Delay between hospitalization and ICU, d	0	(0–1.25)	1	(0–4)	0.96 (0.79–1.16)	0.7
Delay between ICU and IMV, d	0	(0–2)	1	(0–1)	1.18 (0.92–1.51)	0.18

## Data Availability

The data presented in this study are available on request from the corresponding author.

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
