# Peer review of "Ventilator Acquired Pneumonia in COVID-19 ICU Patients: A Retrospective Cohort Study during Pandemia in France"

_jcm, 2023, doi:10.3390/jcm12020421_

Round 1

Reviewer 1 Report

Staphylococcus aureus has demonstrated the ability to interact and infect different cell strains, such as osteoblasts, causing osteomyelitis and bone and joint infections, while becoming increasingly resistant to antibiotic therapy and a reservoir of bacteria that can make infection difficult to treat. please discuss and cite doi:10.3390/pathogens10020239
- The 2019 coronavirus pandemic is a rapidly evolving global emergency that continues to challenge health care systems. Emerging research describes a plethora of patient factors, including demographic, clinical, immunologic, hematologic, biochemical, and radiographic findings, that could be useful to clinicians in predicting the severity and mortality of COVID-19. Current literature related to predictive factors for the clinical course and outcomes of COVID-19 describes findings associated with increased disease severity and/or mortality, including age > 55 years, multiple preexisting comorbidities, hypoxia, specific computed tomography findings indicative of extensive pulmonary involvement, various laboratory test abnormalities, and biomarkers of end-organ dysfunction. please discuss and cite doi:10.1542/peds.2021-053418
- There is a need to better understand the new coronavirus, severe acute respiratory syndrome coronavirus 2 (SARS-CoV-2), for which coronavirus disease 2019 (COVID-19) continues to cause significant morbidity and mortality worldwide. COVID-19 is comparable to other respiratory infectious diseases such as influenza A (H7N9) and influenza A (H1N1) virus infections of avian origin. Patients hospitalized with laboratory-confirmed infection with SARS-CoV-2 (n = 83), H7N9 (n = 36), and H1N1 (n = 44) viruses were analyzed.
Both COVID-19 and H7N9 patients had longer length of hospitalization than H1N1 patients (P < 0.01), higher complication rates, and more severe cases than H1N1 patients. H7N9 patients had a higher hospitalization-to-fatality ratio than COVID-19 patients (P = 0.01). H7N9 patients had similar patterns of lymphopenia, neutrophilia, elevated alanine aminotransferase, C-reactive protein, lactate dehydrogenase and those observed in H1N1 patients, all of which were significantly different from COVID-19 patients (P < 0.01). H7N9 or H1N1 patients had more prominent symptoms, such as fever, fatigue, yellow sputum and myalgia, than COVID-19 patients (P < 0.01). The mean duration of viral shedding was 9.5 days for SARS-CoV-2 versus 9.9 days for H7N9 (P = 0.78). For severe cases, the elapsed time from disease onset to severity was 8.0 days for COVID-19 vs 5.2 days for H7N9 (P < 0.01); the comorbidity of chronic heart disease was more common in COVID-19 patients than in H7N9 (P = 0.02). Multivariate analysis showed that chronic heart disease was a possible risk factor (OR > 1) for COVID-19, compared with H1N1 and H7N9. please discuss and cite doi:10.1093/cid/ciaa1012
- A calixarene derivative (1), with four α-l-C-fucosyl units linked by a flexible spacer, and a monomeric analog (2) with a single fucose moiety were synthesized. Compounds 1 and 2 were tested for antibiofilmic activity against Pseudomonas aeruginosa (Gram-) and Staphylococcus epidermidis (Gram+). Macrocyclic compound 1 showed a very high percentage of biofilm inhibition against two different bacterial strains, while compound 2, which does not possess a macrocyclic structure, showed only moderate biofilm inhibition against P. aeruginosa and no biofilm inhibition against S. epidermidis. Multivalent fucose derivative could be a novel broad-spectrum antibiofilm agent. please discuss and cite doi:10.1016/j.carres.2019.03.005

Author Response

Staphylococcus aureus has demonstrated the ability to interact and infect different cell strains, such as osteoblasts, causing osteomyelitis and bone and joint infections, while becoming increasingly resistant to antibiotic therapy and a reservoir of bacteria that can make infection difficult to treat. please discuss and cite doi:10.3390/pathogens10020239

Discussion of this thematic was inserted at the page 16, with the citation.

- The 2019 coronavirus pandemic is a rapidly evolving global emergency that continues to challenge health care systems. Emerging research describes a plethora of patient factors, including demographic, clinical, immunologic, hematologic, biochemical, and radiographic findings, that could be useful to clinicians in predicting the severity and mortality of COVID-19. Current literature related to predictive factors for the clinical course and outcomes of COVID-19 describes findings associated with increased disease severity and/or mortality, including age > 55 years, multiple preexisting comorbidities, hypoxia, specific computed tomography findings indicative of extensive pulmonary involvement, various laboratory test abnormalities, and biomarkers of end-organ dysfunction. please discuss and cite doi:10.1542/peds.2021-053418

Citation refer to a paediatric population, in our publication an adult population is studded and thus it’s impossible to discuss this paper.

- There is a need to better understand the new coronavirus, severe acute respiratory syndrome coronavirus 2 (SARS-CoV-2), for which coronavirus disease 2019 (COVID-19) continues to cause significant morbidity and mortality worldwide. COVID-19 is comparable to other respiratory infectious diseases such as influenza A (H7N9) and influenza A (H1N1) virus infections of avian origin. Patients hospitalized with laboratory-confirmed infection with SARS-CoV-2 (n = 83), H7N9 (n = 36), and H1N1 (n = 44) viruses were analyzed.
Both COVID-19 and H7N9 patients had longer length of hospitalization than H1N1 patients (P < 0.01), higher complication rates, and more severe cases than H1N1 patients. H7N9 patients had a higher hospitalization-to-fatality ratio than COVID-19 patients (P = 0.01). H7N9 patients had similar patterns of lymphopenia, neutrophilia, elevated alanine aminotransferase, C-reactive protein, lactate dehydrogenase and those observed in H1N1 patients, all of which were significantly different from COVID-19 patients (P < 0.01). H7N9 or H1N1 patients had more prominent symptoms, such as fever, fatigue, yellow sputum and myalgia, than COVID-19 patients (P < 0.01). The mean duration of viral shedding was 9.5 days for SARS-CoV-2 versus 9.9 days for H7N9 (P = 0.78). For severe cases, the elapsed time from disease onset to severity was 8.0 days for COVID-19 vs 5.2 days for H7N9 (P < 0.01); the comorbidity of chronic heart disease was more common in COVID-19 patients than in H7N9 (P = 0.02). Multivariate analysis showed that chronic heart disease was a possible risk factor (OR > 1) for COVID-19, compared with H1N1 and H7N9. please discuss and cite doi:10.1093/cid/ciaa1012

Citation refer to the mortality of COVID-19 ICU patients, in our study, we have a specific interest on VAP in COVID-ICU patients. Thus, we observed the specific mortality on VAP, but not in overall population. The discussion of this paper is not possible to introduce in our paper is not possible because this subject is not aborted.

- A calixarene derivative (1), with four α-l-C-fucosyl units linked by a flexible spacer, and a monomeric analog (2) with a single fucose moiety were synthesized. Compounds 1 and 2 were tested for antibiofilmic activity against Pseudomonas aeruginosa (Gram-) and Staphylococcus epidermidis (Gram+). Macrocyclic compound 1 showed a very high percentage of biofilm inhibition against two different bacterial strains, while compound 2, which does not possess a macrocyclic structure, showed only moderate biofilm inhibition against P. aeruginosa and no biofilm inhibition against S. epidermidis. Multivalent fucose derivative could be a novel broad-spectrum antibiofilm agent. please discuss and cite doi:10.1016/j.carres.2019.03.005

Discussion of this thematic was inserted at the page 14, with the citation.

Reviewer 2 Report

Thank you for giving me an opportunity to check your article titled 'Ventilator Acquired Pneumoia in COVID 19 ICU patients: a retrospective cohort study during pandemic in France'. It's a retrospective study for ventilator associated pneumonia with COVID-19. VAP is the most important and lethal complication during management of COVID-19 in ICU. So, it's an interesting article I think. I understood the risk factors of VAP under COVID-19 are age, sex and pretreatment ICU according to your article. But these factors are common risk factors for severity and prognostic factors for COVID-19 as you know. So, your conclusion is not impressive I think. 

But the causes of bacteria for VAP were very useful, and  very good information regarding drug-resistant organisms.

I have some questions.

1. As mentioned in your article, how should we perform ventilation management to prevent VAP in the future along with COVID-19 treatment, and should prophylactic administration of antimicrobial agents be used? 

2. If prophylactic administration is necessary, should we give HCAP based antimicrobials (TAZ/PIPC etc.)?

3. In immunomodulators of COVID-19 treatment, did you never use remdesivir in standard therapy, however WHO recommended remdesivir for COVID-19 as a standard therapy? And, did Anakinra or Ruxolitinib effective for COVID-19? Please let us know because we never use them.

4. In conclusion, your conclusion is too long, so please make more simply.

This article is just result for management of COVID-19.

I think you need constructive input from your retrospective data for future prevention of VAP.

Author Response

Thank you for giving me an opportunity to check your article titled 'Ventilator Acquired Pneumoia in COVID 19 ICU patients: a retrospective cohort study during pandemic in France'. It's a retrospective study for ventilator associated pneumonia with COVID-19. VAP is the most important and lethal complication during management of COVID-19 in ICU. So, it's an interesting article I think. I understood the risk factors of VAP under COVID-19 are age, sex and pretreatment ICU according to your article. But these factors are common risk factors for severity and prognostic factors for COVID-19 as you know. So, your conclusion is not impressive I think.

But the causes of bacteria for VAP were very useful, and  very good information regarding drug-resistant organisms.

I have some questions.

1. As mentioned in your article, how should we perform ventilation management to prevent VAP in the future along with COVID-19 treatment, and should prophylactic administration of antimicrobial agents be used?

Prophylactic antibiotherapy is not recommended for prevent VAP in ICU patient, COVID and non COVID patient have the same pathophysiological contamination mechanism. Repeated micro inhalation of oro-pharyngeal contaminated secretions. Selective Digestive decontamination may have some interest for oropharynx colonisation but in patient often ventilated for a long time, rational use of antimicrobial therapy is a priority.

2. If prophylactic administration is necessary, should we give HCAP based antimicrobials (TAZ/PIPC etc.)?

Non applicable

3. In immunomodulators of COVID-19 treatment, did you never use remdesivir in standard therapy, however WHO recommended remdesivir for COVID-19 as a standard therapy? And, did Anakinra or Ruxolitinib effective for COVID-19? Please let us know because we never use them.

We don’t use remdesevir for COVID treatment, we try in some patient tozicilizumad with a clinical impression to increasing of nosocomial infection. We stop after one year all adjuvant treatment and just monitoring viral PCR. We manage all COVID patient as standard ARDS with a special attention on post aggressive lung fibrosis.

  1. In conclusion, your conclusion is too long, so please make more simply.

    This article is just result for management of COVID-19.

    I think you need constructive input from your retrospective data for future prevention of VAP.

Conclusion is modified

Round 2

Reviewer 2 Report

Thank you for giving me the opportunity to check your article again as a revised version. I agreed your comments to my questions. I have no more comments to your article.

Author Response

Author's thanks the reviewer for his very interresting comment to improve the quality of manuscrpit